Urbanization alters small rodent community composition but not abundance

Alvarez Guevara Jessica N.
Ball Becky A. becky.ball@asu.edu
School of Mathematical & Natural Sciences, Arizona State University , Glendale , AZ , USA
Ferrenberg Scott
Electronic publication date: 2018 May 30
Publication date: 2018
Volume: 6
Electronic Location ID: e4885
Received 2018 Jan 18; Accepted 2018 May 12
Copyright: ©2018 Alvarez Guevara and Ball
Copyright year: 2018
Copyright holder: Alvarez Guevara and Ball
License: This is an open access article distributed under the terms of the Creative Commons Attribution License, which permits unrestricted use, distribution, reproduction and adaptation in any medium and for any purpose provided that it is properly attributed. For attribution, the original author(s), title, publication source (PeerJ) and either DOI or URL of the article must be cited.
License URL: https://creativecommons.org/licenses/by/4.0/

Keywords: Rodent, Diversity, Abundance, Arid, Urbanization, Arizona, Sonoran desert

Funding: National Science Foundation DEB-1026865 Central Arizona-Phoenix Long-Term Ecological Research (CAP LTER) New College Undergraduate Inquiry and Research Experiences (NCUIRE) This material is based upon work supported by the National Science Foundation under grant number DEB-1026865, Central Arizona-Phoenix Long-Term Ecological Research (CAP LTER). Support for the undergraduate author came from the New College Undergraduate Inquiry and Research Experiences (NCUIRE) program. The funders had no role in study design, data collection and analysis, decision to publish, or preparation of the manuscript.

==============================
Desert ecosystems are one of the fastest urbanizing areas on the planet. This rapid shift has the potential to alter the abundances and species richness of herbivore and plant communities. Herbivores, for example, are expected to be more abundant within urban desert remnant parks located within cities due to anthropogenic activities that concentrate food resources and reduce native predator populations. Despite this assumption, previous research conducted around Phoenix, AZ, USA has shown that top-down herbivory led to equally reduced plant biomass in both urban and outlying locations. It is unclear if this insignificant difference in herbivory at urban and outlying sites is due to unaltered desert herbivore populations or altered activity levels that counteract abundance differences. Small rodent herbivore/granivore populations were surveyed at four sites inside and four sites outside of the core of Phoenix during fall 2014 and spring 2015 in order to determine whether abundances and richness differ significantly between urban and rural sites. In order to survey species composition and abundance at these sites, 100 Sherman traps and eight larger wire traps that are designed to attract and capture small vertebrates such as mice, rats, and squirrels were set at each site for two consecutive trap nights. Results suggest that the commonly assumed effect of urbanization on herbivore abundances does not apply to small rodent populations in a desert city, as overall small rodent abundances were statistically similar regardless of location. Though a significant difference was not found for species richness, a significant difference between small rodent genus richness at these sites was observed, with altered community composition. The compositional differences likely reflect the altered vegetative community and may impact ecological interactions at these sites.

Introduction

Globally, some of the fastest growing populations and most rapidly urbanizing areas are in arid ecosystems (UNDP, 2014). In the Sonoran Desert of the southwestern United States, the Phoenix, AZ metropolitan area is one of the largest and fastest-growing metropolitan areas in the US, with a higher than average population growth for the past several decades (Martin & Stabler, 2002; US Census Bureau, 2015). For example, in the last 25 years, the resident population within the Phoenix metropolitan area has increased by 47% (Davis et al., 2015) to its current 4.6 million people. As a result, the area of undisturbed land within this city alone decreased by 21% from 1985 to 2005 (Buyantuyev, Wu & Gries, 2010).

Urbanization can cause shifts in animal abundances and diversity. Herbivore abundances within urban parks are often expected to be higher than those found in rural areas due to human activities that concentrate food resources and eradicate native predators (Rodewald & Shustack, 2008; Shochat et al., 2010). Trail systems, anthropogenic water sources, surface temperature and the presence of utilities may be favored by certain species to increase their abundance in urban parks (Markovchick-Nicholls et al., 2008; Pianalto & Yool, 2017; Rudd & Bateman, 2015; Switalski & Bateman, 2017). Species richness, on the other hand, is expected to decrease with urbanization (sensu McKinney, 2008; Saari et al., 2016). The homogenization of species in urban areas is often associated with habitat fragmentation and the introduction of non-native species (McKinney, 2006). Additionally, habitat fragmentation and anthropogenic activity can make areas inviable for certain fauna, and can therefore alter their distribution (Markovchick-Nicholls et al., 2008).

Alterations to the community composition of herbivorous organisms can then cause plant communities to shift (Gruner et al., 2008). It is expected that herbivores in urban areas consume and therefore reduce more above-ground plant biomass than those in rural sites due to higher abundances. Despite this assumption, previous research at the Central Arizona-Phoenix Long-Term Ecological Research (CAP-LTER) has shown that herbivory within urban and rural Sonoran Desert remnant parks has led to equally reduced biomass in urban and outlying areas (Davis et al., 2015). It is unclear whether this lack of difference in herbivory is the result of unaltered herbivore populations or different activity levels that counteract differences in population densities, given the lack of published data reporting the abundance of small vertebrate herbivores in and around the Phoenix metropolitan area. As urbanization continues to expand and encroach onto the native land of many herbivores and plants, it is important to study and understand how the lives of these herbivore species, and therefore the plants they eat, are affected.

We quantitatively surveyed small rodent populations at four sites inside (urban) and four sites outside (outlying) of the city core of Phoenix to determine whether abundances and diversity differ significantly with urban activities. Small rodents (including mice, rats, and squirrels) are common vertebrate herbivores, granivores, and omnivores in the Sonoran Desert, which have the potential to impact plant biomass and community composition. We hypothesize that overall abundance of the rodent species found within the desert remnant parks inside the city will be significantly higher, and that the species richness will be significantly lower, than the outlying parks located outside the core of the city of Phoenix.

Materials & Methods

Study site

This study was conducted within remnant parks of the Northern Sonoran Desert of Arizona. All study plots are found within the 6400 km2 CAP LTER boundaries that encompass the area in and around the city of Phoenix (Davis et al., 2015). The average annual rainfall for sites within the core of the city of Phoenix in 2014 & 2015 was 272(±14) mm and 149(±4) mm, respectively, while the average annual rainfall for rural sites was 300(±28) mm and 207(±22) mm in those same years (FCDMC, 2009). Several uncharacteristically large storms in 2014 caused higher precipitation averages than those of previous years, and 2015 was more consistent with the long-term averages (Ball & Alvarez Guevara, 2015; Davis et al., 2015). Dominant plants within the Sonoran Desert ecosystem include creosote (Larrea tridentata), bursage (Ambrosia deltoidea), palo verde (Cercidium spp.), and ironwood (Olneya tesota). Additional plants identified at study cites include succulents, such as saguaros (Carnegiea gigantea), chain fruit cholla (Cylindropuntia fulgida), and teddy bear cholla (Cylindropuntia bigelovii).

Experimental design

Eight desert remnant park sites inside and outside of the city of Phoenix previously utilized for CAP LTER herbivore exclosure studies (Davis et al., 2015) were used in this population census (Fig. 1). In order to directly compare the effect urbanization had on abundance and community composition of small rodents, four urban sites were sampled alongside four outlying sites. Urban sites were located inside of the city of Phoenix, while outlying sites were located to the east of the urban core.

Figure 1 Map of CAP LTER study sites.

Urban sites are those found within the city core, while outlying sites are those found outside of the city core. The urban sites (circles) used in this study consist of Piestewa Peak Park (PWP), Desert Botanical Garden (DBG), and South Mountain Park East (SME) and West (SMW). The outlying sites (triangles) used in this study are Usery Mountain Regional Park (UMP), Lost Dutchman State Park (LDP), Salt River Recreation (SRR), and north McDowell Mountain Regional Park (MCN).

Trapping events took place over four weekends in both the fall (September–October) of 2014 and spring (March–May) of 2015 in order to account for the fluctuation of populations associated with the seasons. These two seasons were selected because small rodents also tend to be most active during spring and fall when extreme heat and cold do not present a mortality concern (Moseley et al., 2011). A single weekend trapping event surveyed both an urban and outlying site for two consecutive nights and mornings. Coupled urban-outlying sites were kept consistent in both the fall and spring, though the order in which the four paired sites were surveyed in the fall were shuffled in the spring to reduce the influence of sampling order on results. Trapping events were not scheduled during full moons, as previous studies indicate that small rodents limit activity in order to reduce exposure to nocturnal predators (Daly et al., 1992). Additionally, trapping events were not scheduled during weekends with severe weather predicted (i.e., thunderstorms or temperature below 40°F) in order to minimize rodent mortality risks. This work was conducted under AZ Game & Fish Scientific Collecting permits SP654186 (2014) and SP694606 (2015) and IACUC protocol #13-1316R at Arizona State University.

Small rodent surveys

Community composition and abundances of mice, rats, and squirrels were quantified using the live capture-release method (Sikes & ACUC, 2016). Traps were scattered within a 20,000 m2 area at each site and placed in key habitat types in order to ensure maximal rodent capture rates. Initial trapping efforts revealed optimal trap placement to be under native plant cover such as palo verde (Cercidium spp.), mesquite (Prosopis spp.), creosote (Larrea tridentata), etc. Desert ecosystems are characterized by a patchy distribution of vascular plants, with exposed interplant spaces between shrubs (Crawford & Gosz, 1982; Schlesinger et al., 1996), and these interplant spaces provide no source of cover or food for small plant-associated rodents (herbivores and granivores in particular) that we were targeting. In fact, preliminary methods testing demonstrated that traps in interplant spaces were almost entirely empty during trapping events. To increase our trapping success and the likelihood of observing maximum numbers of individuals and taxa, we targeted plant-based habitat types across defined, replicated areas of the Sonoran Desert, as described below.

At each site for two consecutive nights, 100 Sherman folding traps (7.62 × 8.89  × 22.86 cm) and 8 larger wire traps (17.78 × 17.78  × 17.78 cm) were set and baited with a mix of rolled oats and toasted oat cereal, totaling 216 traps set per night across the urban-outlying site pairs. Within the set 20,000 m2 area of each study site, four equal quadrants were visualized. Sherman traps 1–25 were placed within quadrant one, Sherman traps 26–50 were placed within quadrant two, etc. Each quadrant also contained two larger wire traps. Traps were set in the late afternoon of the first and second day, and rodents were identified to species the following mornings using Kays and Wilson’s Mammals of North America (Kays & Wilson, 2002).

All traps were closed after the identification of small rodents on the first morning and were kept closed throughout the day until set and baited again that same afternoon. This was done in order to minimize trap mortality associated with the heat of the day. To target diurnal rodents, the order in which the sites were visited on the first morning was reversed on the second morning; this allowed the traps to be open for an extra 2–3 h of daylight while data was being obtained from its paired site.

Data analysis

Small rodent abundance, as well as species and genus richness were analyzed using Analysis of Variance (ANOVA) in R 2.7.2 (The R Foundation) with both Location (urban or rural) and Season (fall or spring) as main effects, as well as their interaction. Data were found to be normal. Eight sites (four urban and four outlying) over two seasons yields 16 total samples. Abundances were defined as the number of rodents captured per 100 trap nights. We do not attempt to calculate density, given that we did not place traps on a random and evenly-spaced grid system. The Shannon Index for diversity (H = ∑pln(p)) and evenness (J′ = H∕Hmax) were calculated and also analyzed using ANOVA. Due to the small replication feasible in this study, we also ran a nonparametric Kruskal-Wallace test on the same data, which yielded the same conclusions as the ANOVA, bolstering the conclusion that are data are normal. A non-metric multidimensional scaling (NMDS) was conducted using the small rodent species abundance data using the metaMDS command in the package “vegan” in R (Oksanen et al., 2018), where stress is 0.0580. A Permutational Multivariate Analysis of Variance (PERMANOVA) was additionally used to test for the impacts of Location*Season on community composition, also using the “vegan” package in R (Oksanen et al., 2018), using the default 999 permutations. It should be noted that several individuals escaped prior to their identification to species, and are therefore only known to the genus level. These data are included in total abundances, but no other metrics or analyses.

Results

Overall, total rodent abundance and species-level diversity did not differ between urban and outlying sites, but did at the genus level. Small rodent abundances were slightly higher within the urban desert remnant parks (Fig. 2A), though this difference was statistically insignificant (Table 1). Measures of diversity at the species level, including species richness, Shannon index, and species evenness, tended to be slightly higher in the outlying sites but were again not significantly different across location (Figs. 2B–2D, Table 1). Variation in species richness within urban parks was higher because the South Mountain sites tended to be more diverse than the Desert Botanical Garden and Piestewa Peak sites (standard error in Fig. 2B). In contrast to the lack of difference in species diversity, outlying sites outside of the city are significantly greater in genus richness than urban sites (Fig. 2E, Table 1). Season and its interaction with location did not significantly influence any of the measures of rodent community (Table 1), so all data discussed are pooled across the year.

Figure 2 Small rodent community characteristics in urban and outlying sites.

Characteristics (n = 4, average ± SE) include (A) total abundance, (B) species richness, (C) Shannon diversity index, (D) species evenness, and (E) genus richness.

Table 1 Results of the Analysis of Variance (ANOVA) analyzing rodent abundance, species and genera richness, and diversity indices, as well as of the Permutational Multivariate Analysis of Variance (PERMANOVA) analyzing rodent community composition, according to Location (urban or outlying), Season (fall or spring), and their interaction.

F statistics are expressed as the value for F, with a subscript of the degrees of freedom (df) of the factor being tested followed by the df for the Error.

	Abundance	Species richness	Shannon index	Species evenness	Genus richness	Community composition	
	F	P	F	P	F	P	F	P	F	P	F	P	
Location	1.9861,12	0.184	0.9821,12	0.341	2.4961,12	0.140	0.3911,12	0.544	16.6151,12	0.002	3.1271,12	0.027	
Season	0.2011,12	0.662	0.0001,12	1.000	0.2281,12	0.642	0.7831,12	0.394	0.0001,12	1.000	2.2031,12	0.061	
Location: Season	1.0361,12	0.329	1.7461,12	0.211	2.1941,12	0.164	1.0841,12	0.318	0.46151,12	0.510	0.9811,12	0.428	

Beyond measures of diversity, community composition differed between urban and outlying locations. The PERMANOVA identified a significant effect of Location on community composition of species (Table 1). Certain taxa were associated with either urban or outlying sites (Table S1). Deer mice (Peromyscus spp.) were only identified within the urban sites, while grasshopper mice (Onychomys spp.) and kangaroo rats (Dipodomys spp.) were only identified at outlying park sites. Further, the NMDS shows a separation of urban and outlying sites, with all outlying parks grouping together in the upper left-to-central portion of the ordination and urban parks on the lower right-to-central portion (Fig. 3). This difference is driven by the higher abundances of certain species at outlying parks, such as white-throated woodrats (Neotoma albigula) and Merriam’s kangaroo rats (Dipodomys merriami; Table S1). Within the outlying sites, the McDowell Mountain Regional Park (MCN) rodent community differed from Usery Mountain Regional Park (UMP) and Salt River Recreation (SRR) in that it contained northern grasshopper mice (Onychomys leucogaster). The Lost Dutchman State Park (LDP) rodent community differed in that it contained a relatively high abundance of Mexican woodrats (Neotoma mexicana) in comparison to UMP and SRR.

Figure 3 Non-metric multidimensional scaling (NMDS) of rodent communities found at urban and rural sites.

Position of sites depends on individual abundances of species indicated by the vectors. The urban sites are Piestewa Peak Park (PWP), Desert Botanical Garden (DBG), and South Mountain Park East (SME) and West (SMW), and the rural sites are Usery Mountain Regional Park (UMP), Lost Dutchman State Park (LDP), Salt River Recreation (SRR), and north McDowell Mountain Regional Park (MCN). Species abbreviations are the first two letters of the genus and species as listed in the full species list in Table S1, with Pg being the abbreviation for Perognathus (pocket mice) and Pm being the abbreviation for Peromyscus (deer mice). Individuals that escaped prior to identification, for whom the genus is known but not the species, were left out of the analysis

The NMDS also shows that the Desert Botanical Garden (DBG) and Piestewa Peak (PWP) differ from the South Mountain West (SMW) and South Mountain East (SME) sites in terms of small rodent species community composition. The Desert Botanical Garden site is mainly composed of Bailey’s (Chaetodipus baileyi) and desert (C. penicillatus) pocket mice (Table S1, Fig. 3), while Piestewa Peak is the only site in which round-tailed ground squirrels (Xerospermophilus tereticaudus) were captured.

Discussion

Small rodent species richness, community composition, and abundance were measured in both urban and outlying desert remnant parks in order to assess the impact of urbanization. We hypothesized that overall small rodent abundances measured within urban desert remnant parks would be higher than overall abundances of small rodents found within outlying desert remnant parks. According to the data, however, the commonly assumed difference in abundance between urban and outlying parks does not apply to small rodent populations in a desert city when manicured environments are excluded. This supports the previous study that did not find a significant difference in aboveground plant biomass consumption when comparing urban and outlying desert remnant parks (Davis et al., 2015). A recent meta-analysis also shows that, across ecosystems and terrestrial animal taxa considered, there is not a general trend of increased abundance with urbanization (Saari et al., 2016). In fact, contrary to the common assumption, the authors found evidence for decreased abundance in urban areas, though this effect became insignificant when outlier European studies involving arthropods were removed. Our data demonstrate that Sonoran Desert rodents are a further example of an ecosystem and taxa that do not fit the generalization that urbanization increases abundance.

We also hypothesized that species richness would be higher in outlying parks, though this was not found to be the case. The insignificant difference between the Shannon Index and evenness support this finding, indicating that the urban and outlying parks sampled have similar levels of diversity and evenness at the species level. These results further support the conclusions of the meta-analysis by Saari et al. (2016), which also showed statistically insignificant differences in species richness with urbanization across studies. However, the meta-analysis did not look at community composition beyond species richness, and in our study genus richness was found to be statistically greater at outlying parks. This, along with the PERMANOVA and NMDS results, means that the community composition of the small vertebrate rodents do differ across site location.

There are many potential mechanisms that would result in the decreased diversity (sensu Saari et al., 2016), including habitat loss in the urban setting. It’s possible that the plant communities upon which small rodents are dependent determine which parks they inhabit. For example, it is possible that rodent richness and diversity is the result of the diversity of the local plant community. Previous studies have shown that these outlying desert remnant parks have a more diverse plant community than urban remnant parks (Davis et al., 2015). Though both urban and outlying desert parks are dominated by Curvenut Combseed (Pectocarya recurvata), Arabian Schismus (Schismus arabicus), and Indian Plantago (Plantago ovata), the average plant percent coverage of these species is higher at outlying parks (Davis et al., 2015). Higher percent coverage of certain shrubs may lead to higher small rodent abundances, as they can provide both food and shelter (Tietje, Lee & Vreeland, 2008).

The significant difference of genus richness observed between urban and outlying sites highlights that pocket mice genera dominate urban sites, specifically Bailey’s (Chaetodipus baileyi), desert (C. penicillatus), rock (C. intermedius), and Arizona (Perognathus amplus) pocket mice. Of these rodents, Bailey’s, rock, and desert pocket mice are classified under the genus Chaetodipus. This may indicate homogenization of small rodents within urban parks, as closely related species tend to have similar ecological roles (Cavender-Bares et al., 2009). For example, the desert pocket mouse is known to larder hoard, which involves caching their food resources in a single burrow. Merriam’s Kangaroo Rats, on the other hand, are known to scatter hoard their food resources in shallow pits (Leaver & Daly, 2001). These different behaviors could influence plant communities, given that the caching behavior of pocket mice, for example, can influence seed germination and invasive grass establishment (Sommers & Chesson, 2016; Walker, Vrooman & Thompson, 2015).

Some species were not identified at the outlying sites used in this study, though it is important to note that these species may be present at these outlying sites at low abundances. These species include the White-Footed (Peromyscus leucopus), North American (P. maniculatus), and Cactus Deer Mice (P. eremicus). Previous studies have shown that this genus may be able to flourish in urban desert remnant parks because their population dynamics are not significantly affected by the moderate removal of shrub cover and food resources, indicating that they may not have a preference in terms of storing food in the open or under shrub cover (Parmenter & MacMahon, 1983).

Similarly, certain species were only identified at outlying sites. Merriam’s Kangaroo Rats (Dipodomys merriami), for example, were found in all four outlying sites, but were not identified at any of the urban sites. Merriam’s kangaroo rats have been found to prefer to pilfer seed caches located under shrub cover (Swartz, Jenkins & Dochtermann, 2010). This may be why they were found at higher abundances within sites that contain higher plant percent coverage. Previous studies suggest that removal of kangaroo rat species (Dipodomys spp.) leads to a significant increase in abundances of other seed-eating rodents such as pocket mice (Chaetodipus and Perognathus spp.) and deer mice (Peromyscus spp.) species due to decreased interspecific competition pressures (Brown & Munger, 1985) and can influence plant communities (Curtin et al., 2000).

Northern Grasshopper Mice (Onychomys leucogaster) were only identified within the outlying McDowell Mountain site. These species, like pocket and deer mice, are not significantly affected by the moderate removal of plant cover (Parmenter & MacMahon, 1983). The location of this species may therefore be dependent on additional vital resources offered within the McDowell site. Grasshopper Mice differ from the other mice identified in that their diet almost exclusively consists of arthropods, especially during the summer months (Hope & Parmenter, 2007). According to the diversity-trophic structure hypothesis, arthropod richness is influenced by plant richness (Knops et al., 1999). This may therefore contribute to a higher richness of insects of particular import to grasshopper mice survival at outlying parks.

The intermediate disturbance hypothesis is often used to explain differences in abundance and diversity between disturbed urban and undisturbed locations. However, our data did not fully support the hypothesis, given that abundance did not differ between our urban and outlying sites. In this study, outlying sites were defined as areas with lower levels of disturbance when compared to urban sites that were located within the city of Phoenix. Our outlying sites are of low-to-intermediate levels of anthropogenic disturbance, which in comparison to other truly undisturbed sites would be expected to have higher levels of abundance. As such, abundances measured at the much less disturbed Cave Creek Bajada were found to be lower than those observed within the rural parks of this study (Brown & Zeng, 1989). Another possible reason our results were not in line with generalized patterns like the intermediate disturbance hypothesis is that our sampling design of targeting particular habitats for trap placement biased our results towards plant-associated species of interest. This bias was replicated across locations, so should not compromise the observed differences between locations. It is possible, though, that the intermediate disturbance hypothesis would be supported if we had surveyed the entire population, rather than plant-associated species.

In summary, neither small rodent abundance nor species richness differed significantly between urban and outlying desert remnant parks in this study. Genus richness, however, was found to be significantly higher within the outlying sites, indicating that small rodent community composition differs between these urban and outlying sites. It is important to further research the impacts small rodents can have on desert remnant parks and the plant communities within. Certain activities of these species have been linked to an increase in biodiversity and landscape heterogeneity (Davidson & Lightfoot, 2006). Food storage mounds and underground tunnels made by Banner-tailed Kangaroo Rats, for example, can lead to nitrogen and phosphate rich patches of soil that are preferred habitat for some desert plants (Eldridge, Whitford & Duval, 2009). The presence of small rodents can therefore be important indicators of the health of a desert remnant park.

Conclusion

In sum, our results suggest that the commonly assumed effect of urbanization on herbivore abundances does not apply to small rodent populations in this desert city, as overall small rodent abundances were statistically similar regardless of location. Urban activity did, however, influence community composition and diversity. Though a significant difference was not found for species richness, a significant difference between small rodent genus richness at these sites was observed, and certain taxa were specifically associated with either urban or outlying locations. The compositional differences likely reflect the altered vegetative community and may impact ecological interactions at these sites.

Supplemental Information

Table S1 Abundance (# individuals/100 trap nights) of each species identified in each of the sites surveyed, organized by genus

The urban sites used in this study consist of Piestewa Peak Park (PWP), Desert Botanical Garden (DBG), and South Mountain Park East (SME) and West (SMW). The rural sites are Usery Mountain Regional Park (UMP), Lost Dutchman State Park (LDP), Salt River Recreation (SRR), and north McDowell Mountain Regional Park (MCN).

Click here for additional data file.

We thank the AZG&F Dept. for lending us traps during our preliminary research, and the management team for each of the city, regional, and state parks used for access to the sites. Lastly, we would like to give a special thanks to the following family, friends, and volunteers that provided invaluable assistance in data collection: Miguel A. Alvarez, Maria M. Alvarez Guevara, Miguel Alvarez Guevara, Martha Alvarez, Kelly Bergin, Sean Hosier, Miranda Vega, Connor Wetzel-Brown, Ivan Fernandez, Jordan Patterson, Lexi Lake, Gabie Santos, Yesenia Rodriguez-Sanchez, Sungho Park, Cesar Ruiz, Uffe Nielsen, Marcia Denmark, and Paige Smith.

Additional Information and Declarations

Competing Interests

Author Contributions

Animal Ethics

Field Study Permissions

Data Availability

The authors declare there are no competing interests.

Jessica N. Alvarez Guevara conceived and designed the experiments, performed the experiments, analyzed the data, prepared figures and/or tables, authored or reviewed drafts of the paper, approved the final draft.

Becky A. Ball conceived and designed the experiments, analyzed the data, contributed reagents/materials/analysis tools, prepared figures and/or tables, authored or reviewed drafts of the paper, approved the final draft.

The following information was supplied relating to ethical approvals (i.e., approving body and any reference numbers):

Arizona State University IACUC provided approval for this research (IACUC protocol #13-1316R).

The following information was supplied relating to field study approvals (i.e., approving body and any reference numbers):

Field experiments were approved by Arizona Game & Fish (Scientific Collecting permits SP654186 (2014) and SP694606 (2015)).

The following information was supplied regarding data availability:

Ball B. 2018. Impact of urbanization on small rodent abundance and community composition. Environmental Data Initiative. http://dx.doi.org/10.6073/pasta/a8ce3e859f142c58a89baa0dd54bd6d3.

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
