# Peer review of "Urbanization alters small rodent community composition but not abundance"

_PeerJ, doi:10.7717/peerj.4885_

## Round 0.1 · original submission · Major Revisions

Two expert reviewers have considered this manuscript and offered a number of useful suggestions that will improve the quality of this submission. I recommend considering all of the suggested changes from both reviewers. However, of these suggestions and critiques, the primary issues that I would like to see addressed relate to concerns raised by Reviewer 1. Specifically, please address Reviewer 1's concerns regarding sampling effort and potential bias. This is a potentially serious design issue and more explanation of how the traps were dispersed and why is needed before a decision can be made.

In addition, I agree with Reviewer 1 that small mammals that were identified to different taxonomic levels (i.e., species vs. genus when unknown) is an issue when using an NMDS to assess your data. If these unknown species can retroactively be parsed to reliable "morphospecies" treating them as consistent sets of unknowns would be a better option than lumping several unknowns within a genus before computing a distance matrix and plotting an NMDS. I would also recommend considering an analytical approach to comparing communities, in addition to the graphical view of NMDS, such as a PERMANOVA or ANOSIM. This type of analysis can include 'strata' allows you to retain the full value of species richness and abundances in a manner that cannot be matched with diversity indices.

Reviewer 1 ·

Basic reporting

In general, the manuscript is well-written. However, there is some imprecise and inaccurate use of terminology. For example, throughout the manuscript, instead of referring to the study organisms as ‘small mammals’ or ‘rodents,’ the authors often refer to ‘herbivores’ and ‘small vertebrates’ – both of these terms encompass many species other than those studied here. But, more importantly, many of the small mammals in this study are not actually herbivores, but rather granivores (for example, Dipodomys spp.) or omnivores (Peromyscus spp.).

Figure 1 needs a scale bar. Currently, it is impossible to tell how far apart the sites are.

In the Figure 2 caption, n should equal 4, not 8, because there are 4 sites/treatment.

There are several places in the manuscript where the writing should be tightened or clarified:

Lines 22-23: This sentence is unclear – “led to equally reduced plant biomass” – as compared to what?

Line 61: The use of “distribution” here is awkward. I’m not sure that the distribution (do you mean range?) of a species is necessarily reduced by anthropogenic effects, although the distribution could certainly change as a result of human influence.

Lines 62-63: This is an incomplete sentence.

Line 67: Again, what does “equally reduced biomass” mean?

Line 72: The reference to “vertebrates and plants” strikes me as odd. It leaves out invertebrates entirely, and then the sentence goes on to refer to “the plants they eat.” This is really unclear, because as written, the sentence implies that plants eat plants.

Line 151: It is unclear what is meant by “abundance” here – are the authors referring to abundance by individual species, or to total small mammal abundance including multiple species?

Experimental design

The authors investigate an important and timely question: how urbanization affects populations of native animals. Unfortunately, the experimental design is flawed, and the sampling strategy does not actually yield the data needed to test their hypothesis. The critical issue, and one that can’t be fixed at this point, is that traps were not distributed across the study areas in an unbiased way (lines 121-122). To be able to make a valid comparison of small mammal presence and abundance across sites, traps should either have been randomly distributed (for example, by generating random X,Y points and placing traps at those points), or in a consistent pattern such as a trap grid, web, or line. By “scattering” traps and placing them in locations where animals were more likely to be captured, the authors biased their data. This approach is fine if the only objective is to capture as many animals as possible (perhaps for behavioral studies of a particular species, for example), but completely inappropriate if the goal is to determine the species and numbers of animals present in an unbiased way. By selectively placing traps in certain types of microhabitats, the authors have potentially overestimated the abundance of certain species and missed others entirely.

There are also concerns with the statistical analysis. The authors did test for normality of the data, but with such small sample sizes (n of 4 in each category), such a test is of limited utility. I would suggest the use of non-parametric tests instead. I am also concerned with the NMDS, because of the way in which “unknown” animals and individuals that were classified to genus but not species were included. Inspection of Fig. 3 reveals that these categories of animals are de facto treated as separate species. This means that there are 5 “species” included in this analysis that are not real:UK, ChSpp, PmSpp, NeSpp, and OnSpp. In effect, this takes individuals from multiple actual species within the genus and lumps them together, which is problematic in multiple ways.

The authors did have the appropriate permits and protocols, but need to include the university that approved the IACUC protocol (I assume ASU, but this is not stated).

There are much more recent guidelines for the care and use of mammals in research from the American Society of Mammalogists – I believe the most recent set was published in 2016, and is freely available on the ASM website.

Validity of the findings

Unfortunately, the data were collected in such a way that they cannot actually be used to test the hypothesis posed in this manuscript. The authors could reanalyze their data in more appropriate ways (such as removing the animals that were not identified to species from the NMDS), but because of the bias in the trapping strategy and therefore the underlying data, valid comparisons between urban and outlying areas cannot be made.

·

Basic reporting

Very good, no problems

Experimental design

Very good, no problems

Validity of the findings

Very good, no problems

Additional comments

A very nice contribution to the literature, well written and well implemented. My only suggestion is that you cite and add a discussion of the article below, which is a meta-analysis of papers documenting abundance changes in urban habitats. Your paper should be placed in the context of their results: does it fit? If not, then why? Also, they discuss mechanisms of abundance change: how do your results compare to their suggested mechanisms?

Urbanization is not associated with increased abundance or decreased richness of terrestrial animals - dissecting the literature through meta-analysis
By:Saari, S (Saari, Susanna)[ 1 ] ; Richter, S (Richter, Scott)[ 2 ] ; Higgins, M (Higgins, Michael)[ 2 ] ; Oberhofer, M (Oberhofer, Martina)[ 3 ] ; Jennings, A (Jennings, Andrew)[ 3 ] ; Faeth, SH (Faeth, Stanley H.)[ 3 ]
URBAN ECOSYSTEMS Volume: 19 Issue: 3 Pages: 1251-1264 DOI: 10.1007/s11252-016-0549-x
Published: SEP 2016 Abstract
The widely accepted consensus is that urbanization increases abundance but reduces species richness of animals. This assumption is the premise for empirical tests and theoretical explanations. We studied the association of urbanization with abundance and species richness of different animal taxa in 20 and 26 published articles reporting abundances and richness, respectively via meta-analysis. Because some articles had multiple estimates, we analyzed 40 and 58 estimates of abundance and richness, respectively. Contrary to conventional wisdom, the overall abundance of terrestrial animals was not higher in urban areas, but instead actually lower, while we failed to confirm the conventional thinking of lower species richness with urbanization. These findings cannot, however, be generalized across all cities and animal species, as conflicting differences were reported among geographical regions, animal taxa. Our results question the conventional wisdom that urbanization generally increases abundances while reducing species richness, and highlights the variability of urbanization effects on diversity among taxa and geographic regions.

---

## Round 0.2 · Major Revisions

I am pleased to see your attempts to respond to all reviewer comments from the first round of reviews. However, in my effort to assess whether your manuscript needs further review, I have noted some issues that must be resolved before I can proceed with a decision. In particular, there is an issue with the analysis and figures that need attention. In your rebuttal letter, you note that in Figure 2 the mistake of listing n=8 has been corrected to reflect the true replication number as n = 4. Yet I see that figure 2 still has n=8 in the legend. More concerning to me, however, is that in Table 1 the degrees of freedom for all factors is listed as 1,12. It is unclear how an experiment with 2 location types (urban vs. outlying) and 2 seasons (fall vs. spring) can have a denominator d.f.=12 for any factor. Thus, either the experimental design is not accurately explained, or the analysis was performed in a way that gave an incorrect output. This next part is speculation, but I see that analyses were performed in R and one important check is to make sure that fixed factors such as location and season are not being viewed as numeric--an issue that occurs when locations or treatments are given number identifiers instead of text identifiers which is something many of us do in data recording. R will still perform the ANOVA even if the factors are viewed as numeric but the output will be incorrect. If this might be the case, you can check with the command "is.numeric(Location)".

While I primarily scanned the methods and results before catching the issues noted above, I caught one other small type on Line 77 where outlying is not enclosed by parentheses on both sides.

Please check your experimental design description, replication descriptions, and analyses and resubmit. I will then assess the manuscript and make a decision.

---

## Round 0.3 · accepted · Accept

Thank you for detailing your study design in response to my earlier comments. The design is now clear to me and the changes will eliminate confusion for other readers as well. Thank you for choosing PeerJ for your work. Please note that PeerJ does not offer copy editing so please be sure to thoroughly check your forthcoming proofs to catch any potential errors.

#